# Botanical Oils Isolated from *Simmondsia chinensis* and *Rosmarinus officinalis* Cultivated in Northern Egypt: Chemical Composition and Insecticidal Activity against *Sitophilus oryzae* (L.) and *Tribolium castaneum* (Herbst)

**DOI:** 10.3390/molecules27144383

**Published:** 2022-07-08

**Authors:** Rady Shawer, Mohamed Mahrous El-Shazly, Adel Mohamed Khider, Rowida S. Baeshen, Wafaa M. Hikal, Ahmed Mohamed Kordy

**Affiliations:** 1Department of Plant Protection, Faculty of Agriculture (Saba Basha), University of Alexandria, Alexandria 21531, Egypt; elshazly@alexu.edu.eg (M.M.E.-S.); dradelkhider@gmail.com (A.M.K.); ahmedkordy@alexu.edu.eg (A.M.K.); 2Department of Biology, Faculty of Science, University of Tabuk, Tabuk 71421, Saudi Arabia; rbaeshen@ut.edu.sa (R.S.B.); wafaahikal@gmail.com (W.M.H.); 3Environmental Parasitology Laboratory, Water Pollution Research Department, Environment and Climate Change Institute, National Research Centre (NRC), Giza 12622, Egypt

**Keywords:** stored-product pest, toxicity, jojoba, rosemary, botanical oil, bioassay, IPM

## Abstract

The rice weevil, *Sitophilus oryzae* (L.), and the red flour beetle, *Tribolium castaneum* (Herbst), are key stored-product pests in Egypt and worldwide. The extensive use of synthetic insecticides has led to adverse effects on the environment, human health, and pest resistance. As a result, environmentally friendly pest management alternatives are desperately required. The botanical oils of jojoba, *Simmondsia chinensis* (L.)*,* and rosemary, *Rosmarinus officinalis* L. plants growing in Egypt were extracted, identified by gas chromatography/mass spectrometry (GC–MS), and evaluated for their insecticidal activity against *S. oryzae* and *T. castaneum*. The main constituents identified in BOs were carvyl acetate (20.73%) and retinol (16.75%) for *S. chinensis* and camphor (15.57%), coumarin (15.19%), verbenone (14.82%), and 1,8-cineole (6.76%) for *R. officinalis.* The *S. chinensis* and *R. officinalis* BOs caused significant contact toxicities against *S. oryzae* and *T. castaneum* adults, providing LC_50_ values of 24.37, 68.47, and 11.58, 141.8 ppm at 3 days after treatment (DAT), respectively. *S. chinensis* oil exhibited significant fumigation toxicity against both insects; however, it was more effective against *S. oryzae* (LC_50_ = 29.52 ppm/L air) than against *T. castaneum* (LC_50_ = 113.47 ppm/L air) at 3 DAT. Although the essential oil (EO) of *R. officinalis* significantly showed fumigation toxicity for *S. oryzae* (LC_50_ = 256.1 and 0.028 ppm/L air at 1 and 3 DAT, respectively)*,* it was not effective against *T. castaneum*. These BOs could be beneficial for establishing IPM programs for suppressing *S. oryzae* and *T. castaneum*.

## 1. Introduction

Globally, insect pests cause significant losses in stored products post-harvest annually [1]. In less-developed countries, insect damage to stored food grains is estimated at 10–40% [2]. Insects are key pests for stored crops, where they attack their seed embryos and negatively affect germination [3,4]. In Egypt, the losses in wheat grains due to insect injury were estimated at 35–55% [5,6]. *Sitophilus oryzae* (L.) and *Tribolium castaneum* (Herbst) are major pests that cause significant damage to grain-based products during the storage process [1,7,8,9]. The chemical control tool is the main method to manage these stored-product pests. However, the widespread use of synthetic insecticides raises serious concerns about insect resistance, residues on grains, and increased negative environmental effects. As a result, finding eco-friendly approaches has become an urgent need [10,11]. Natural and non-persistent insecticides are the most important priorities in this regard [1,12,13,14]. The botanical oils [15] extracted from plants have favorable ecotoxicological properties, including low toxicity to humans, further degradation, biodegradability, and lower environmental impact [16,17,18,19,20]. They can be toxic to various important insect pests as fumigants [21,22], contact [23], repellent [23,24,25], antioxidant [26], or antifeedant [27,28,29]. The efficacy of several EOs extracted from aromatic plants has been widely evaluated for the control of such pests [27,28] and has exhibited promising results [29,30].

*Rosmarinus officinalis* L. (Lamiales: Lamiaceae) is a medicinal plant native to Europe, but it has been cultivated in different areas around the world [31,32,33]. The EO of *R. officinalis* has been widely used in medicine due to its powerful antibacterial, cytotoxic, antimutagenic, antioxidant, antiphlogistic, and chemopreventive characteristics [34,35]. In a trial conducted in Argentina, the EO of *R. officinalis* was found to be highly effective on *Tribolium castaneum* [36]. The jojoba plant, *Simmondsia chinensis* (L.) (Caryophyllales: Simmondsiaceae), is a monotypic species native to the Sonoran Desert in North America [37]. The oil of jojoba seeds is a mixture of long-chain esters (97–98%) of fatty acids and fatty alcohols. It has been reported that the oil has significant analgesic, antipyretic, anti-inflammatory, antioxidant, anti-bacterial, and anti-parasitic properties [38]. Furthermore, it has a lethal effect on the adults of *S. oryzae* (L.) [39].

The bioactivity of these BOs is affected by their chemical profiles [40,41,42,43,44]. Furthermore, the environmental conditions of different regions impact the chemical constituents of these plants [35,40]. It has been reported that BOs collected from different growing areas at different periods have different chemical compositions and therefore may exhibit different biological activities [35,41]. However, while many studies have been conducted in different countries to assess the insecticidal performance of various BOs against stored-product insects, the available data on BOs extracted from plants growing in Egypt and their bioactivity against common stored-product pests are scarce. Therefore, in the present study, we extracted *S. chinensis* and *R. officinalis* BOs from plants cultivated in Northern Egypt and identified their chemical composition by GC–MS. In addition, the toxicity of those isolated BOs against the rice weevil, *S. oryzae*, and the red flour beetle, *T. castaneum*, was studied under laboratory conditions.

## 2. Results

### 2.1. Chemical Composition of Botanical Oils

The yielded oils of *S. chinensis* and *R. officinalis* from the extraction process were 40.57 and 0.71% v/w, respectively. The percentages of chemical constituents identified in *S. chinensis* and *R. officinalis* BOs are summarized in Table 1. Fourteen compounds were identified in *S. chinensis* BO, representing 99.98% of total constituents. The major components were carvyl acetate (20.73%), retinol (16.75%), gibberellic acid (15.34%), igernellin (7.12%), and retinal (5.52%). A total of 42 compounds were characterized in *R. officinalis* EO, representing 99.02% of total constituents. The major identified component was camphor (15.57%), followed by coumarin (15.19%), verbenone (14.82%), 1,8-cineole (6.76%), hymecromone (5.56%), and α-Pinene (4.29%).

### 2.2. Contact Bioassay

Mortality of *S. oryzae* and *T. castaneum* adults caused by the contact bioassay of different concentrations of *S. chinensis* BO at 1, 2, and 3 DAT is shown in Table 2. All the evaluated concentrations of *S. chinensis* oil significantly reduced the number of treated *S. oryzae* adults at 1, 2, and 3 DAT in comparison with controls. The most effective activity was caused by using a concentration of 200 ppm of *S. chinensis* oil, showing 90, 93.3, and 96.7% mortality at 1, 2, and 3 DAT, respectively. The same trend was repeated against *T. castaneum* adults, except for concentrations of 10 and 20 ppm, which were not significantly different compared to control at 1 and 2 DAT. The concentration of 200 ppm showed the best activity at 1, 2, and 3 DAT. It is observed that mortality of both insects treated with *S. chinensis* BO was increased when oil concentration and/or exposure time were increased. The estimated contact LC_50_ values were 54.35, 36.94, and 24.37 ppm/cm^2^ for *S. oryzae*, whereas in the case of *T. castaneum* they were 181.3, 101.5, and 68.47 ppm/cm^2^, respectively, at 1, 2, and 3 DAT (Table 3).

The treatments of high concentrations (≥50 ppm/cm^2^) of *R. officinalis* EO showed significant contact toxicities of *S. oryzae* adults higher than control at 1 DAT (Table 4). Moreover, all the evaluated concentrations significantly decreased the number of *S. oryzae* adults at 2 and 3 DAT in comparison with control. The lowest two concentrations of *R. officinalis* EO (10 and 20 ppm/cm^2^) were not able to cause a reduction in the number of adults *T. castaneum* at 1, 2, or 3 DAT, showing no mortality. The highest concentration (200 ppm/cm^2^) of *R. officinalis* EO showed significant activity against *T. castaneum* adults, providing 40, 53.3, and 67% mortality, respectively, at 1, 2, and 3 DAT. The LC_50_ values of *R. officinalis* EO recorded on *S. oryzae* were 115.8, 40.54, and 11.58 ppm/cm^2^ at 1, 2, and 3 DAT, respectively (Table 5). The values decreased by about a third every day of exposure time. At 1, 2, and 3 DAT, the LC_50_ values against *T. castaneum* were 281.9, 213.7, and 141.8 ppm/cm^2^, respectively. The LC_50_ values recorded against *T. castaneum* were higher than those against *S. oryzae*.

### 2.3. Fumigation Bioassay

The results of the fumigation bioassay of *S. chinensis* BO confirmed that all treatments significantly reduced the number of both adults *S. oryzae* and *T. castaneum* at 1, 2, and 3 DAT compared to control (Table 6). However, the oil was more effective against *S. oryzae* than *T. castaneum*. The efficacy of *S. chinensis* BO against both insects following the fumigant bioassay was improved while exposure time was increased. The greatest activity against *S. oryzae* and *T. castaneum* adults was caused by the highest concentration (200 ppm/L air) of *S. chinensis* BO, showing 68.3 and 27% mortality at 3 DAT, respectively. The LC_50_ values of *S. chinensis* BO recorded on *S. oryzae* and *T. castaneum* adults following the fumigation bioassay were 159.1, 202.7, 29.52, and 179.59, 168.46, 113.47 ppm/L air, respectively, at 1, 2, and 3 DAT (Table 7). At 3 DAT, the LC_50_ value in the case of *S. oryzae* was decreased about five times more than it was at 1 DAT.

According to the results of the fumigation bioassay, *R. officinalis* EO was effective against adult *S. oryzae* but was not on *T. castaneum* (Table 8 and Table 9). In comparison with control, all treatments caused significant mortality of *S. oryzae* adults at 1, 2, and 3 DAT. The mortality of *S. oryzae* adults was in the ranges 30–52%, 54–85%, and 78–95%, respectively, at 1, 2, and 3 DAT. Mortality of *S. oryzae* increased exponentially while *R. officinalis* oil concentration was increasing. *R. officinalis* fumes showed zero activity against *T. castaneum*, since all tested concentrations were unable to cause any adult mortality. The LC_50_ values recorded by *R. officinalis* oil against *S. oryzae* adults were 256.1 ppm/L air at 1 DAT (Table 9). The LC_50_ value was highly reduced at 2 and 3 DAT, showing 9.4 and 0.028 ppm/L air, respectively.

## 3. Discussion

Different previous studies have documented the insecticidal characteristics of many plant-derived substances that would enable them to play an effective role in the integrated pest management (IPM) of stored-product pests. These substances may be advantageous because they are selective for pests and have no or little negative impact on non-target organisms or the environment [1,27]. Furthermore, many of these products are biodegradable by soil microorganisms and are low in toxicity to mammals [42]. BOs and their major constituents are complementary tools for pest control, in particular of stored-product insects [23]. They can be used in IPM of stored-product insects for pest prevention, early pest detection, or pest control [15,43,44]. In the present study, the BOs of *S. chinensis* and *R. officinalis* were extracted from local plants growing in Egypt, identified by GC–MS, and evaluated against *S. oryzae* and *T. castaneum* by contact and fumigation bioassays under laboratory conditions. The chemical compositions of the extracts from *R. officinalis* and *S. chinensis* BOs were similar to those previously reported for the oils isolated from plants cultivated in Egypt [44] and other countries around the world [31,42,45,46,47,48,49,50]. However, the percentages of constituents differed. This can be attributed to many factors, including the difference in geographical site, collection period, environmental and climate conditions, and the nutritional status of the plants [44,51]. In 2016, Abdelgaleil et. al. extracted the EO of *R. officinalis* growing in the same area of plant collection as the present study (Northern Egypt) [44]. The main common constituents identified in both studies were 1,8-cineole, camphor, a-pinene, verbenone, and L-linalool. Moreover, in the current study, we report the presence of coumarin (15.19%) and hymecromone (5.56%) in the *R. officinalis* EO. However, additional research and information are ideally required to support our findings regarding the presence of those compounds. These two compounds have previously been found in plant extracts such as *Orysan sativa* [52]. The variation of the active ingredients within a plant extract would improve its mechanism of action and thus its biological effectiveness [53]. It is reported that linalool has an acetylcholinesterase inhibition and repellence effect against some insects [54,55]. Furthermore, the presence of terpenes and aromatic volatile compounds in the BOs had anti-bacterial and anti-fungal effects and protected food products for a long time without affecting their quality [54,56]. The 1.8-cineole was found to be highly effective against adults of *S. oryzae* when it was applied at a concentration of 0.1 mL/720 mL volume at 24 h of exposure, whereas camphor was found to be more effective towards *Rhyzopertha dominica*, with mortality of 100% [57,58]. The GC–MS analysis revealed that the major component found in *S. chinensis* BO was carvyl acetate (20.73%). It is a natural product identified in different plant sources such as citrus, *Mentha pulegium*, and *Santolina chamaecyparissus* [59].

In this study, both oils of *S. chinensis* and *R. officinalis* exhibited different degrees of toxicity against *S. oryzae* and *T. castaneum*. These results are in harmony with those previously found by Abdel-Rahman and Mahmoud [60], who observed high contact activity of *S. chinensis* oil against *S. oryzae* adults. The estimated LC_50_ and LC_90_ values of *S. chinensis* against *S. oryzae* were 1.17 and 2.76% (*v*/*v*), respectively, at 4 DAT. In a trial conducted in Egypt, the repellent and toxicant effects of eight BOs on *T. castaneum* adults were studied [54]. The oil of *S. chinensis* provided 73.33% adult repellency at 180 min after exposure at a concentration of 15% (*v*/*v*). The contact toxicity of the *S. chinensis* BO at the same concentration showed a relatively low LC_50_ value (10.73% *v*/*v*) at 1 DAT [54].

However, the degree of toxicity depended on the bioassay method [44] and the targeted insect. For example, *S. chinensis* BO was effectively excellent against *S. oryzae* (90% mortality) and moderate against *T. castaneum* (70% mortality) when it was used in the contact bioassay at 1 DAT. In addition, the *S. chinensis* and *R. officinalis* BOs were significantly effective as contact toxicants against both *S. oryzae* and *T. castaneum* adults. However, both were more effective against *S. oryzae* than on *T. castaneum* adults. While *S. chinensis* oil exhibited significant fumigation toxicity against *S. oryzae* (LC_50_ = 29.52 ppm/L air), higher than that on *T. castaneum* (LC_50_ = 1134.7 ppm/L air), the *R. officinalis* EO had a moderate effect on *S. oryzae* (LC_50_ = 256.1 ppm/L air at 1 DAT) and was ineffective against *T. castaneum*. Those results are in harmony with those obtained by Saroukolai et al. (2010), who found that the extracted *Thymus persicus* EO was 70 times more effective against *S. oryzae* than *T. castaneum* adults [1]. In different studies, *R. officinalis* EO showed strong fumigation toxicity against *S. oryzae*. The estimated LC_50_ values varied from 26.71 mg/L air to 53.6 286 µL/L air [29,60]. Similarly, *Origanum vulgare* EO was found to be an efficacious fumigant rather than a contact against *S. oryzae* [44]. Furthermore, *A. monosperma* and *P. graveolens* had high contact toxicity against *S. oryzae* but were ineffective fumigants [44]. On the other hand, the oil of *Cupressus sempervirens* caused strong fumigant toxicity for *S. oryzae* but was not effective in the contact method [61]. In summary, the results of the biological performance of these botanical oils is affected by application methods [62,63], and this should be considered when developing IPM programs for such insects.

## 4. Materials and Methods

### 4.1. Insect Colonies

*S.**oryzae* and *T. castaneum* insects used in the current experiment originated from a susceptible strain that was previously reared in the laboratory of the Plant Protection Department, Faculty of Agriculture (Saba-Basha), Alexandria University, Egypt. For *S. oryzae* rearing, four hundred adults (mixed sexes and ages) were placed into 2 L glass jars containing 500 g of sterilized wheat grains (var. Sakha 68) [5,6,64]. To avoid the escape of insects, about 7–10 cm from the inside upper part of the used jars were painted with Vaseline. For well ventilation, jars were covered with pieces of muslin fixed to the neck of the jars by two rubber bands [65]. After one week, all the added adults were removed using 8-mesh sieves, and the culture was kept in a hygrothermal conditioned cabinet (28 ± 2 °C and 70 ± 5% R.H.). The culture was thereafter investigated daily, and the emerged adults were used in bioassays. These procedures were performed to obtain groups of adults of the same and known age. All the same procedures and conditions used with *S. oryzae* were followed in *T. castaneum* rearing except the used medium (wheat flour, bran, and dry yeast at a rate of 17:5:1, respectively). Adult insects used in the current experiment were 2–3 weeks old [43].

### 4.2. Extraction of Botanical Oils

The seeds of jojoba, *Simmondsia chinensis* (L.), were collected from a farm located in Al-Adl village, Gharb Al-Nubaria region, Nubaria city, Al-Behieira Governorate, Egypt (30°39′51.9″ N, 30°07′28.0″ E). Healthy jojoba seeds were selected and transferred to the laboratory of the Plant Protection Department, Faculty of Agriculture (Saba Basha), Alexandria University in plastic bags. The BO from seeds was extracted by the pressing method [66]. The seeds were well dried for 14 days. The oil extraction process was made at the room temperature, using a manual hydraulic press (SPECAC, London, UK) with a load configuration of 0–5 tones. About 50 g of seeds was added to the press cylinder and subjected to the pressing process. The yielded oil (*v*/*w*%) was determined and then saved for further chemical analysis and bioassays. For the extraction of *R. officinalis* EO, fresh leaves were collected from a known nursery located in the Abees region, Alexandria governorate, Egypt, and transferred in plastic bags to the laboratory. Leaves were dried at room temperature (26 ± 1 °C) for five days and were subjected to hydro-distillation using a Clevenger apparatus for 6 h [67,68,69,70,71]. The resulting oil was filtered, dried over anhydrous sodium sulfate, expressed as *v*/*w*% of the dry matter [44], and stored in the laboratory refrigerator at 4 °C until usage.

### 4.3. Chemical Composition of Botanical Oils

The chemical composition of BOs was performed using a Thermo-Logical Gas Chromatography (GC Follow 1300)/Mass Spectrometer (ISQ7000 show; Thermo Logical) apparatus (Agilent Technologies, Santa Clara, CA, USA) [47,62]. A Thermo TR-50MS capillary column (30 m in length × 250 µm in breadth × 0.25 µm in thickness of film) was used as a GC column. The spectroscopic location in GC–MS included an electron ionization framework that used high-energy electrons (70 eV) and a 300 °C MS exchange line temperature. Unadulterated helium gas (99.995%) was used as the carrier gas with a flow rate of 1 mL/min. The column temperature was programmed (60 °C for 2 min, 100 °C at 10 °C/min for 5 min, 150 °C at 10 °C/min for 5 min, 200 °C at 10 °C/min for 5 min, and 250 °C at 10 °C/min for 20 min). One microliter of the arranged extricates was infused in a partless mode.

### 4.4. Contact Bioassay

The contact activity of six serial concentrations (10, 20, 50, 100, 150, and 200 ppm) of both *S. chinensis* and *R. officinalis* BOs was evaluated against the adults of *S.*
*Oryzae* and *T. castaneum* with the method previously described [29]. The BOs were diluted in acetone (Al-Nasr Pharmaceutical Chemicals Co., Obour, Egypt). One mL of each concentration was placed in a 9 cm Petri dish by a micropipette and spread uniformly on the whole surface of the dish. Acetone was allowed to evaporate, leaving a thin film of the oils on the surface of dishes [41]. Twenty adults (same age and weight) of each insect were separately added to the Petri dish. Dishes without BOs acted as controls. All treatments were replicated five times. At 1, 2, and 3 days after treatment (DAT), the percentages of adult mortality were recorded and corrected using Abbott’s formula [1,72].
$$\mathrm{Corrected}\;\mathrm{mortality}\%=\frac{\left(\mathrm{Mortality}\%\;\mathrm{of}\;\mathrm{treated}\;\mathrm{insects}-\mathrm{Mortality}\%\;\mathrm{of}\;\mathrm{control}\right)}{\left(100-\mathrm{Mortality}\%\;\mathrm{of}\;\mathrm{control}\right)}\times 100$$

The LC_50_ values (concentration causing 50% mortality compared with the control) expressed as ppm/cm^2^ were calculated [73].

### 4.5. Fumigation Bioassay

To evaluate the fumigation toxicity of BOs against *S. oryzae* and *T. castaneum* adults, six oil concentrations (10, 20, 50, 100, 150, and 200 ppm/L air) were evaluated following the previously described bioassay [29,74]. One-liter glass jars were used as fumigation chambers. Each oil concentration was evenly added to a filter paper piece (2 × 3 cm) fixed in the subsurface of the screw caps of jars. The inner side of the jar’s neck was painted with Vaseline to prevent direct contact of insects with the treated filter paper. Caps were directly screwed tightly onto the jars, each containing 20 adults (same age and weight). The filter papers in the controls were treated with acetone only. Each treatment was replicated five times. The adult mortality was calculated at 1, 2, and 3 DAT, and the LC_50_ values (ppm/L air) were considered [73].

### 4.6. Statistical Analysis

The generalized linear model (GLM) was used to perform a one-way analysis of variance on the insect mortality data [32]. Means were then compared by the Duncan’s least significant difference (LSD) test [75] using SAS software V. 8.2 (SAS Institute Inc., Cary, NC, USA) [76]. Differences were considered significant at α = 0.05. The LdP line computerized software program was used to calculate the probit analyses of LC_50_ values and their fiducial limits (confidence intervals) for botanical oils according to Finney (1971) [77].

## 5. Conclusions

Botanical oils have been widely investigated for their biological activity against a wide range of agricultural pests, including stored-product pests. Many studies have confirmed the potency of these products; however, their toxicity is dependent on a number of factors (e.g., the chemical composition of the BO, the targeted pest, and the bioassay method). The data of the current study suggest the high contact and fumigation effects of *S. chinensis* and *R. officinalis* BOs against *S. oryzae* and *T. castaneum*. Thus, it can be concluded that these BOs are a promising approach in terms of decreasing chemical pesticide use, and they should be considered for an effective IPM strategy for *S. oryzae* and *T. castaneum*.

## Figures and Tables

**Table 1 molecules-27-04383-t001:** Chemical compositions identified in *Rosmarinus officinalis* and *Simmondsia chinensis* botanical oils.

RT ^1^	RI ^2^	Compound Name	Concentration (%)	
			*S. chinensis*	*R. officinalis*
5.20	930	α-Pinene	-	4.29
5.73	935	Camphene	-	0.98
7.28	1014	Limonene	-	0.76
7.91	1023	1,8-Cineole	-	6.76
9.75	1082	Linalool	-	1.40
10.15	2122	Linolenic acid	4.36	-
12.77	1170	Verbenone	-	0.62
13.56	1146	Camphor	-	15.57
14.03	1172	α-Terpineol	-	1.86
14.17	1148	Borneol	-	0.74
14.33	1156	3-Pinanone	-	0.78
14.86	1266	Thymol	-	0.45
15.28	2112	Methyl 2,5-octadecadiynoate	3.31	-
15.56	1264	2,5-Bornanediol	-	0.80
15.69	1170	Verbenone	-	14.82
16.28	2495	Androstanolone	-	0.32
17.31	1273	Ascaridole	-	0.92
17.80	1407.76	Caryophyllene	-	1.61
18.22	1206	Linalyl formate	-	0.59
19.44	1275	Carvacrol	-	0.91
20.12	1421	β-Caryophyllene	-	0.84
21.38	2466	Retinal	-	0.51
22.40	2000	Falcarinol	-	0.45
22.98	NA ^3^	Picrotoxin	-	0.64
23.39	1386	Cinnamic acid	-	0.55
23.53	1488	Butylated hydroxytoluene	-	0.71
24.02	2466	Retinal	5.52	-
24.83	3131	Campesterol	-	1.69
25.00	1414	Coumarin	-	15.19
25.28	849	13Z,16Z-docosadienoic acid	-	0.42
25.54	2112	Methyl 2,5-octadecadiynoate	-	1.76
25.85	1575	Caryophyllene oxide	-	2.47
26.22	2301	methyl (E)-heptadec-10-en-8-ynoate	-	0.41
26.66	1629	Methyl jasmonate	-	0.70
27.08	2832	1,12-Di(oxiran-2-yl)dodecane	-	0.77
28.18	2102	linolenic acid	-	0.56
28.96	2112	13,16-Octadecadiynoic acid methyl ester	-	0.41
30.51	NA	Bakuchiol	-	1.39
31.52	1345	Carvyl acetate	20.73	-
32.07	2003	Hymecromone	-	5.56
32.74	3942	1-Heptatriacotanol	-	1.52
33.5	NA	2,5-Octadecadiynoic acid, methyl ester	2.65	-
34.73	2034	Falcarinol	-	0.80
36.1	2451	Retinol	16.75	-
37.10	2735	6beta-Naltrexol	-	2.47
37.11	2843	dihydrotachysterol	3.93	-
37.24	2393	Gibberellic acid	-	0.98
39.04	2285	Dihydroxanthin	2.91	-
40.05	1988	Ethylene brassylate	5.13	-
40.30	1831	tert-Hexadecanethiol	-	1.51
40.48	2451	Retinol	-	2.95
40.84	NA	Igernellin	7.12	-
43.18	2122	alpha-Linolenic acid	2.91	-
43.26	NA	Gibberellic acid	15.34	-
46.06	2151	Isofetamid	3.26	-
47.91	NA	Martynoside	2.13	-
Total identified			99.98	99.02

^1^ RT, Retention time; ^2^ RI, Retention index as determined on a TR-50MS capillary column; ^3^ NA, Not available in device database for used column.

**Table 2 molecules-27-04383-t002:** Mortality (±SD) of *S. oryzae* and *T. castaneum* adults following contact bioassay of *S. chinensis* oil.

Conc. ^1^ (ppm/cm^2^)	Mortality of *S. oryzae*			Mortality of *T. castaneum*		
	Days after Treatment			Days after Treatment		
	1	2	3	1	2	3
10	15.0 ± 2.65 ^e^*	23.3 ± 1.35 ^d^	32.0 ± 1.65 ^e^	9.00 ± 1.15 ^de^	13.3 ± 0.58 ^de^	17.0 ± 1.15 ^e^
20	25.0 ± 1.00 ^e^	35.0 ± 3.00 ^d^	46.7 ± 1.65 ^d^	12.0 ± 0.58 ^de^	14.0 ± 0.58 ^de^	25.0 ± 1.53 ^de^
50	40.0 ± 2.00 ^d^	51.7 ± 1.65 ^c^	61.7 ± 1.65 ^c^	22.0 ± 0.65 ^cd^	28.3 ± 0.35 ^cd^	34.0 ± 2.52 ^d^
100	61.7 ± 1.65 ^c^	69.0 ± 2.35 ^b^	77.0 ± 2.35 ^b^	30.0 ± 3.00 ^c^	39.0 ± 0.35 ^c^	53.3 ± 2.35 ^c^
150	76.7 ± 2.65 ^b^	81.7 ± 1.35 ^ab^	88.3 ± 1.65 ^ab^	51.7 ± 1.65 ^b^	57.0 ± 1.65 ^b^	71.0 ± 2.00 ^b^
200	90.0 ± 1.00 ^a^	93.3 ± 0.35 ^a^	96.7 ± 0.35 ^a^	70.0 ± 2.00 ^a^	82.0 ± 1.35 ^a^	87.0 ± 1.35 ^a^
Control	0.00 ± 0.00 ^f^	0.00 ± 0.00 ^e^	0.00 ± 0.00 ^f^	0.00 ± 0.00 ^e^	0.00 ± 0.00 ^e^	0.00 ± 0.00 ^f^

^1^ Conc., oil concentration; * Means within each column followed by the same letter(s) are not significantly different (Duncan’s LSD test; *p* = 0.05).

**Table 3 molecules-27-04383-t003:** The LC_50_ values of *S. chinensis* oil recorded against *S. oryzae* and *T. castaneum* adults following contact bioassay.

Insects	DAT ^1^	LC_50_ ^2^ (ppm/cm^2^)	95% Confidence Limits (ppm/cm^2^)		Slope ^3^ ± SE	(X^2^) ^4^
			Lower	Upper		
*S. oryzae*	1	54.35	46.20	63.58	1.63 ± 0.13	9.22
	2	36.94	30.42	44.04	1.48 ± 0.13	7.97
	3	24.37	19.28	29.59	1.45 ± 0.13	7.02
*T. castaneum*	1	181.3	99.21	205.4	1.39 ± 0.19	24.7
	2	101.5	69.88	165.9	1.82 ± 0.15	13.9
	3	68.47	46.08	103.3	1.71 ± 0.14	12.2

^1^ DAT, Days after treatment; ^2^ LC_50_, the concentration causing 50% mortality; ^3^ Slope of the concentration inhibition regression line ± SE; ^4^ Chi square value.

**Table 4 molecules-27-04383-t004:** Mortality (±SD) of *S. oryzae* and *T. castaneum* adults following contact bioassay of *R. officinalis* oil.

Conc. ^1^ (ppm/ cm^2^)	Mortality of *S. oryzae*			Mortality of *T. castaneum*		
	Days after Treatment			Days after Treatment		
	1	2	3	1	2	3
10	4.00 ± 0.58 ^de^*	24.0 ± 1.35 ^d^	50.0 ± 4.00 ^b^	0.00 ± 0.00 ^c^	0.00 ± 0.00 ^d^	0.00 ± 0.00 ^d^
20	13.3 ± 0.58 ^cde^	39.0 ± 0.35 ^cd^	60.0 ± 1.65 ^b^	0.00 ± 0.00 ^c^	0.00 ± 0.00 ^d^	0.00 ± 0.00 ^d^
50	23.3 ± 1.35 ^cd^	50.0 ± 4.00 ^bc^	73.3 ± 3.34 ^ab^	6.67 ± 1.15 ^c^	13.3 ± 0.58 ^cd^	21.7 ± 0.65 ^c^
100	35.0 ± 2.52 ^bc^	57.0 ± 1.65 ^bc^	77.0 ± 2.35 ^ab^	17.0 ± 2.65 ^b^	22.0 ± 0.65 ^bc^	31.7 ± 3.00 ^c^
150	53.3 ± 2.35 ^b^	73.3 ± 3.34 ^ab^	90.0 ± 1.00 ^a^	25.0 ± 1.53 ^b^	34.0 ± 2.52 ^b^	50.0 ± 4.00 ^b^
200	79.0 ± 2.35 ^a^	94.0 ± 0.35 ^a^	95.0 ± 1.00 ^a^	40.0 ± 1.70 ^a^	53.3 ± 2.35 ^a^	67.0 ± 1.40 ^a^
Control	0.00 ± 0.00 e	0.00 ± 0.00 e	0.00 ± 0.00 c	0.00 ± 0.00 ^c^	0.00 ± 0.00 ^d^	0.00 ± 0.00 ^d^

^1^ Conc., oil concentration; * Means within each column followed by the same letter(s) are not significantly different (Duncan’s LSD test; *p* = 0.05).

**Table 5 molecules-27-04383-t005:** The LC_50_ values of *R. officinalis* oil recorded against *S. oryzae* and *T. castaneum* adults following contact bioassay.

Insects	DAT ^1^	LC_50_ ^2^ (ppm/cm^2^)	95% Confidence Limits (ppm/cm^2^)		Slope ^3^ ± SE	(X^2^) ^4^
			Lower	Upper		
*S. oryzae*	1	115.8	78.47	212.5	1.73 ± 0.15	20.2
	2	40.54	15.84	75.67	1.27 ± 0.12	6.82
	3	11.58	7.127	16.16	1.06 ± 0.13	1.04
*T. castaneum*	1	281.9	141.8	444.4	2.12 ± 0.38	075
	2	213.7	175.8	297.0	1.97 ± 0.33	3.14
	3	141.8	422.7	169.9	2.06 ± 0.31	3.95

^1^ DAT, Days after treatment; ^2^ LC_50_, the concentration causing 50% mortality; ^3^ Slope of the concentration inhibition regression line ± SE; ^4^ Chi square value.

**Table 6 molecules-27-04383-t006:** Mortality (±SD) of *S. oryzae* and *T. castaneum* adults following the fumigation bioassay of *S. chinensis* oil.

Conc. ^1^ (ppm/L Air)	Mortality of *S. oryzae*			Mortality of *T. castaneum*		
	Days after Treatment			Days after Treatment		
	1	2	3	1	2	3
10	5.00 ± 1.00 ^bc^*	27.0 ± 1.53 ^ab^	44.0 ± 2.00 ^a^	0.00 ± 0.00 ^b^	1.67 ± 0.58 ^b^	4.00 ± 0.58 ^de^
20	12.0 ± 0.58 ^abc^	33.3 ± 2.51 ^a^	48.3 ± 1.35 ^a^	0.00 ± 0.00 ^b^	2.00 ± 0.58 ^b^	7.00 ± 1.15 ^cde^
50	12.0 ± 0.58 ^abc^	35.0 ± 2.52 ^a^	49.0 ± 1.65 ^a^	3.33 ± 0.58 ^b^	8.33 ± 1.15 ^ab^	12.0 ± 0.58 ^bcd^
100	18.3 ± 3.21 ^abc^	44.0 ± 2.00 ^a^	57.0 ± 1.65 ^a^	5.00 ± 1.00 ^b^	9.00 ± 1.15 ^ab^	19.0 ± 3.21 ^abc^
150	21.7 ± 0.65 ^ab^	47.0 ± 1.35 ^a^	62.0 ± 1.65 ^a^	8.33 ± 1.15 ^b^	12.0 ± 0.58 ^ab^	22.0 ± 0.65 ^ab^
200	30.0 ± 3.00 ^a^	52.0 ± 2.35 ^a^	68.3 ± 1.40 ^a^	17.0 ± 2.65 ^a^	19.0 ± 3.21 ^a^	27.0 ± 1.53 ^a^
Control	0.00 ± 0.00 ^d^	0.00 ± 0.00 ^c^	0.00 ± 0.00 ^b^	0.00 ± 0.00 ^c^	0.00 ± 0.00 ^c^	0.00 ± 0.00 ^f^

^1^ Conc., oil concentration; * Means within each column followed by the same letter(s) are not significantly different (Duncan’s LSD test; *p* = 0.05).

**Table 7 molecules-27-04383-t007:** The LC_50_ values of *S. chinensis* oil against *S. oryzae* and *T. castaneum* adults following the fumigation bioassay.

Insects	DAT ^1^	LC_50_ ^2^ (ppm/L Air)	95% Confidence Limits (ppm/L Air)		Slope ^3^ ± SE	(X^2^) ^4^
			Lower	Upper		
*S. oryzae*	1	159.1	61.53	138.21	0.71 ± 0.15	3.08
	2	202.7	113.6	712.3	0.48 ± 0.14	0.84
	3	29.52	11.30	50.73	0.43 ± 0.11	2.05
*T. castaneum*	1	179.59	71.18	199.22	0.27 ± 0.17	3.50
	2	168.46	69.86	186.93	0.21 ± 0.02	1.744
	3	113.47	57.49	128.11	0.15 ± 0.18	1.41

^1^ DAT, Days after treatment; ^2^ LC_50_, the concentration causing 50% mortality; ^3^ Slope of the concentration–inhibition regression line ± SE; ^4^ Chi square value.

**Table 8 molecules-27-04383-t008:** Mortality (±SD) of *S. oryzae* and *T. castaneum* adults following the fumigation bioassay of *R. officinalis* oil.

Conc. ^1^ (ppm/L Air)	Mortality of *S. oryzae*			Mortality of *T. castaneum*		
	Days after Treatment			Days after Treatment		
	1	2	3	1	2	3
10	30.0 ± 3.00 ^b^*	54.0 ± 1.65 ^a^	78.3 ± 2.35 ^b^	0.00 ± 0.00 ^a^	0.00 ± 0.00 ^a^	0.00 ± 0.00 ^a^
20	30.0 ± 3.00 ^b^	58.3 ± 1.65 ^a^	84.0 ± 0.85 ^ab^	0.00 ± 0.00 ^a^	0.00 ± 0.00 ^a^	0.00 ± 0.00 ^a^
50	37.0 ± 2.52^b^	63.3 ± 1.40 ^a^	90.0 ± 1.00 ^ab^	0.00 ± 0.00 ^a^	0.00 ± 0.00 ^a^	0.00 ± 0.00 ^a^
100	41.7 ± 1.35^ab^	74.0 ± 3.34 ^a^	92.0 ± 0.35 ^a^	0.00 ± 0.00 ^a^	0.00 ± 0.00 ^a^	0.00 ± 0.00 ^a^
150	46.7 ± 1.35 ^ab^	78.3 ± 2.35 ^a^	95.0 ± 1.00 ^a^	0.00 ± 0.00 ^a^	0.00 ± 0.00 ^a^	0.00 ± 0.00 ^a^
200	51.7 ± 1.65 ^a^	85.0 ± 0.85 ^a^	95.0 ± 1.00 ^a^	0.00 ± 0.00 ^a^	0.00 ± 0.00 ^a^	0.00 ± 0.00 ^a^
Control	0.00 ± 0.00 ^c^	0.00 ± 0.00 ^b^	0.00 ± 0.00 ^c^	0.00 ± 0.00 ^a^	0.00 ± 0.00 ^a^	0.00 ± 0.00 ^a^

^1^ Conc., oil concentration; * Means within each column followed by the same letter(s) are not significantly different (Duncan’s LSD test; *p* = 0.05).

**Table 9 molecules-27-04383-t009:** LC_50_ values of *R. officinalis* oil recorded against *S. oryzae* and *T. castaneum* adults following fumigation bioassay.

Insects	DAT ^1^	LC_50_ ^2^ (ppm/L Air)	95% Confidence Limits (ppm/L Air)		Slope ^3^ ± SE	(X^2^) ^4^
			Lower	Upper		
*S. oryzae*	1	256.1	128.2	1560	0.42 ± 0.11	1.04
	2	9.404	3.497	16.3	0.66 ± 0.12	11.6
	3	0.028	0.024	0.062	0.34 ± 0.14	11.6

^1^ DAT, Days after treatment; ^2^ LC_50_, the concentration causing 50% mortality; ^3^ Slope of the concentration inhibition regression line ± SE; ^4^ Chi square value.

## Data Availability

Not applicable.

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
