# Peer review of "Botanical Oils Isolated from Simmondsia chinensis and Rosmarinus officinalis Cultivated in Northern Egypt: Chemical Composition and Insecticidal Activity against Sitophilus oryzae (L.) and Tribolium castaneum (Herbst)"

_molecules, 2022, doi:10.3390/molecules27144383_

Round 1
Reviewer 1 Report
Dear Authors,
The manuscript is interseting and well prepared. However, the Materials and Methods section followed by nomenclature should be revised. When given pressing of jojoba seeds, the fraction that you had obtained is not an essential oil but fatty oil. This fraction, when separated in typical for EO, GC-MS conditions, showed a high percentage of e.g. retinol, giberelic acid, which are non volatile terpenic compounds coexisting in fatty oil with fatty acids. So, it is highly recomended to change the nomenclature throught whole text by changing in the case of jojoba: fatty oil or lipid fraction instead of essential oil. It would be good to complete analysis into FA composition.
Reviewer 2 Report
Check the references: line 65 - (Bruneton, 1999)
Line 231: T. castaneum in italic
Line 254-255: About 50 g of seeds were added to the press cylinder and subjected to the pressing process. How many ml of oil you obtained in this process?
Same question for Rosmarinus officinalis
Reviewer 3 Report
Generally, the authors present an interesting study. However, there are some problems in this study and this paper may be considered for publication with major correction. Please find my specific comments below:
1. Line 89, Table 1, I would suggest the authors present the RI (Retention Index) value for each identified compound in Table 1. Only RT value was not enough to identify each compound.
2. Line 89, Table 1, because the authors did not use chiral column, D or L compound cannot be distinguished, for D-Limonene and D-Verbenone, D should be removed.
3. Line 107, Table 3, 0.0.13 should be 0.13, 0.0.19 should be 0.19.
4. Line 144, Table 6, Line 161, Table 8, please double check the letters for the control.
5. Line 186-189, the authors reported a very high content of coumarin (15.19%), however, other researchers did not reported this result in the same area. It is highly impossible that the same plant EOs extracted in the same area had such huge difference. Because the authors of this paper did not use RI values to identify each compound, this assignment probably was wrong (this compound may not be coumarin). Please double check it.
6. Line 246, how much extraction yield was obtained for each EO?
7. Line 262, 4C should be 4°C.
Reviewer 4 Report
The manuscript entitled "Essential Oils Isolated from Simmondsia chinensis and Rosmarinus officinalis Cultivated in Northern Egypt: Chemical Composition and Insecticidal Activity Against Sitophilus oryzae (L.) and Tribolium castaneum (Herbst)" is a good piece of work done by the authors and it is scientifically sound and also contributing new informations to the science. However, there should be some clarity in certain areas; please find my comments and try to incorporate these details in the manuscript during revision. The manuscript may be considered for publication after a major revision
1. The abstract should have a good background section describing the need of pest control and the significance of green pesticides
2. The IC50 values doesn't show any standard error/ deviation
3. The key words should be improved and replaced with new one
4. The references 10-13 are seems to a misfit to the paragraph; I recommed to modify/include the references with the following ones;
https://doi.org/10.3390/pr9071243
https://doi.org/10.1016/j.envres.2021.111718
https://doi.org/10.1007/s10340-009-0261-1
https://doi.org/10.3389/fagro.2022.876687
https://doi.org/10.3390/insects11120836
5. Reference 15 may be replaced with new ones as it is more fit to the statement
https://doi.org/10.1016/j.arabjc.2021.103482
https://doi.org/10.3390/insects13050480
https://doi.org/10.1007/s11356-018-2068-1
6. Line no. 65, the reference (Bruneton, 1999) is not given according to the journal format and neither included in reference list.
7. How many times the GCMS analysis was repeated. There is not mean±SD values.
8. It will more appreciated if the authors can carry out the effect of these essential oils on seed germination (wheat, rice or any other). It will enhace the value of the article (especially as a green pesticide in the stores of seeds)
9. Line no. 218 citation style is wrong
10. Is there any information on the biosafety aspects of these essential oils? Include these details in the discussion to improve the utility of the essential oil as a biologically safe green pesticide.
Round 2
Reviewer 1 Report
The manuscript was properly corrected and can be published now.
Reviewer 3 Report
The authors have tried hard to adjust the text to be clearer. Most questions raised by the reviewers are well answered. I believe the manuscript has been significantly improved. However, there is still one question need to be resolved.
I have suggested the authors present the RI (Retention Index) value for each compound in Table 1 because only RT value (with mass data) was not enough to identify each compound. The authors cited some papers that had only RT values in the reply letter. In my opinion, for an essential oil, if its composition was well studied by many researchers and all the compounds were well known, and if the new paper did not find obvious difference in comparison with former papers, it is ok to use RT value only. Otherwise, it is necessary to measure RI value to further proof the name assignment. The authors of this paper changed coumarin to 2H-1-Benzopyran-2-one in the new manuscript, however, they are the same compound. Based the Mass spectra the authors provided, it likely the name assignment was correct. I would suggest the authors do one of the following three things: 1. Measure and present the RI (Retention Index) value for each compound in Table 1, which will make this paper a better quality. 2. Buy the standard coumarin sample, run GC-MS at the same conditions to further proof it. 3. IF the authors have difficulty to add above experiment, they should mention this issue in the main text that the name of this compound was attempted assigned, further data was needed to proof it.
Overall, I suggest this paper need a minor correction to be published in Molecules.

Reviewer 4 Report
Authors have improved the overall quality of the manuscript and hence I recommend the acceptance of the manuscript. However, some minor errors are there that can be rectified during proof correction;
1. Page 2 line 62-63 citations not listed in reference section and not as per format
2. Page 2 Line 66-67 citations not listed in reference and not as per format
3. Page 8 Line 251- 252 citations not in format and list in reference
Author Response
Please see the attachment

This manuscript is a resubmission of an earlier submission. The following is a list of the peer review reports and author responses from that submission.